# The *in vivo* hydrocarbon formation by vanadium nitrogenase follows a secondary metabolic pathway

Johannes G. Rebelein[1], Chi Chung Lee[1], Yilin Hu[1] & Markus W. Ribbe[1,2]

The vanadium (V)-nitrogenase of *Azotobacter vinelandii* catalyses the *in vitro* conversion of carbon monoxide (CO) to hydrocarbons. Here we show that an *A. vinelandii* strain expressing the V-nitrogenase is capable of *in vivo* reduction of CO to ethylene ($C_2H_4$), ethane ($C_2H_6$) and propane ($C_3H_8$). Moreover, we demonstrate that CO is not used as a carbon source for cell growth, being instead reduced to hydrocarbons in a secondary metabolic pathway. These findings suggest a possible role of the ancient nitrogenase as an evolutionary link between the carbon and nitrogen cycles on Earth and establish a solid foundation for biotechnological adaptation of a whole-cell approach to recycling carbon wastes into hydrocarbon products. Thus, this study has several repercussions for evolution-, environment- and energy-related areas.

[1] Department of Molecular Biology and Biochemistry, University of California, 2236 McGaugh Hall, Irvine, California 92697-3900, USA. [2] Department of Chemistry, University of California, Irvine, California 92697-2025, USA. Correspondence and requests for materials should be addressed to Y.H. (email: yilinh@uci.edu) or to M.W.R. (email: mribbe@uci.edu).

Nitrogenase catalyses the ambient reduction of nitrogen ($N_2$) to ammonia ($NH_3$), a key step in the global nitrogen cycle[1–3]. Recently, this enzyme was shown to reduce carbon monoxide (CO) to hydrocarbons *in vitro*, an important reaction of potential significance for future development of strategies to recycle the toxic CO gas—a waste product of steel, polyvinyl chloride and ferroalloys industries—into useful chemical and fuel products, such as ethylene ($C_2H_4$), ethane ($C_2H_6$) and propane ($C_3H_8$) (refs 4,5). The 'conventional' molybdenum (Mo)- and 'alternative' vanadium (V)-nitrogenases[6] are two homologous members of this enzyme family. Both enzymes comprise a reductase component (*nifH*- or *vnfH*-encoded Fe protein) and a catalytic component (*nifDK*-encoded MoFe protein or *vnfDGK*-encoded VFe protein). Catalysis by both nitrogenases involves the formation of a functional complex between their respective component proteins, which facilitates ATP-dependent transfer of electrons from the reductase component to the cofactor site of the catalytic component, where substrate reduction takes place. Interestingly, the V-nitrogenase of *Azotobacter vinelandii* is considerably more active than its Mo-counterpart in CO-reduction, catalyzing this reaction at a rate of 16 nmol reduced carbon per nmol protein per min as compared to a rate of 0.02 nmol reduced carbon per nmol protein per min by the Mo-nitrogenase[5]. This observation has led to questions of whether the cells expressing V-nitrogenase can also reduce CO to hydrocarbons *in vivo* and, if so, whether these cells incorporate the CO-derived carbon into the cell mass or if they simply release products of CO-reduction as byproducts.

Here we show that the V-nitrogenase of *A. vinelandii* is capable of *in vivo* reduction of CO to $C_2H_4$, $C_2H_6$ and $C_3H_8$ via a secondary metabolic pathway. The ability of V-nitrogenase to reduce both CO and $N_2$ *in vivo* points to a plausible evolutionary link between the carbon and nitrogen cycles on Earth, whereas the observation of an increase of product yield with intermittent aeration of cultures incubated with CO, as well as the inability of cells to use CO as a carbon source, suggests the possibility to develop a whole-cell approach to recycling carbon wastes into hydrocarbon products.

## Results

**Hydrocarbon formation by cultures expressing V-nitrogenase.** To examine the ability of nitrogenase to perform *in vivo* CO reduction, *A. vinelandii* strains carrying the encoding genes for the Mo- and V-nitrogenases, respectively, were first grown in 250 ml growth media supplemented with ammonia, an externally supplied nitrogen source, which suppressed the expression of nitrogenase while allowing accumulation of cell mass (Fig. 1a). The growth of the two cultures started to plateau after 20 and 23 h, respectively, indicating a depletion of ammonia in the growth media that served as a signal to turn on the expression of Mo- and V-nitrogenases in the respective cultures. At this point, the culture flasks were capped by airtight stoppers and CO was added at 10% to the gas phases of these nitrogenase-expressing cultures (Fig. 1a, arrows). The cultures were then allowed to grow in the absence of ammonia, while being monitored for the *in vivo* hydrocarbon formation through an hourly analysis of the headspace of each culture by gas chromatography (GC). Excitingly, the culture expressing the V-nitrogenase was capable of *in vivo* production of 1,390 nmol $C_2H_4$, 63 nmol $C_2H_6$ and 8 nmol $C_3H_6$, respectively, over a time period of 8 h (Fig. 1b, blue). GC–mass spectrometry (GC–MS) analysis further demonstrated mass shifts of $+2$, $+2$ and $+3$, respectively, of products $C_2H_4$, $C_2H_6$ and $C_3H_6$ on substitution of $^{13}CO$ for $^{12}CO$, confirming CO as the carbon source of these hydrocarbon products (Fig. 1c,d). The culture expressing the Mo-nitrogenase, on the other hand, was

unable to generate detectable amounts of hydrocarbon products under the same *in vivo* conditions (Fig. 1b, black), consistent with the previous observation that the Mo-nitrogenase was only 0.1% as active as the V-nitrogenase in the *in vitro* reaction of CO reduction[5].

Such an impaired ability of Mo-nitrogenase to reduce CO to hydrocarbons *in vivo* was further demonstrated by subjecting the culture expressing the Mo-nitrogenase to different amounts of CO (between 0 and 37.5% at a step-increase of 7.5%), where no hydrocarbons were detected at any CO concentration (Fig. 2a). In contrast, the culture expressing the V-nitrogenase displayed activities of *in vivo* hydrocarbon formation when CO was supplied at all tested concentrations, reaching a maximum of 1,556 nmol products per 250 ml culture at 15% CO (Fig. 2a). Quantification of V-nitrogenase expressed in the culture further revealed formation of a maximum of 750 nmol reduced carbon per nmol VFe protein, or a turnover number of as high as 750, at 15% CO over 8 h (Fig. 2b). Importantly, CO could not be reduced to hydrocarbons when the expression of V-nitrogenase was suppressed by the addition of excess ammonia in the growth media (Fig. 2a). Moreover, the culture expressing the V-nitrogenase closely resembled the purified V-nitrogenase in the distribution of products[5], producing $C_2H_4$ as the overwhelmingly predominant product (95%; Fig. 2a, black) of CO reduction over $C_2H_6$ (3.9%; Fig. 2a, red) and $C_3H_8$ (1.1%; Fig. 2a, turquoise). Together, these observations established a direct link between the V-nitrogenase in the culture and the *in vivo* activity of hydrocarbon formation.

**Increase of hydrocarbon yield by aeration of cultures.** The observation of good turnover numbers of the *in vivo* CO reduction compelled us to further explore the possibility to improve the yield of hydrocarbon production in this reaction. Noticeably, the *in vivo* production of hydrocarbons by V-nitrogenase started to plateau after the cell culture was incubated with CO for 4 h (see Fig. 1b). In addition, there was a decline of *in vivo* hydrocarbon formation when CO was supplied at a concentration beyond ∼15% (see Fig. 2a). These results could be explained by inhibition of the respiratory chain and/or other key metabolic pathways of *A. vinelandii* by CO, as well as accumulation of toxic waste products and/or inhibitors of V-nitrogenase on CO reduction, leading to an inability of cells and/or the V-nitrogenase to function normally with prolonged exposure to high concentrations of CO. To alleviate the inhibitory effect of CO, the culture expressing V-nitrogenase was placed under air after incubation with 15% CO for 4 h, permitting the cells to 'relax' for 20 min before 15% CO was re-introduced into the gas phase of the culture. Remarkably, such a treatment led to a complete revitalization of the culture in its ability to produce hydrocarbons, as the culture displayed a linear increase of product formation after nearly 20 repetitions of this procedure (Fig. 2c). The amounts of $C_2H_4$, $C_2H_6$ and $C_3H_8$ accumulated after 20 repetitions were 15,680, 625 and 71 nmol per 250 ml culture, respectively (Fig. 2c), demonstrating the possibility of biotechnological adaptation of this protocol for *in vivo* hydrocarbon production in the future. Moreover, a dramatic increase of CO consumption was achieved by this procedure, providing a necessary tool for the determination of whether CO was used as a carbon source and incorporated into cell mass or whether it was processed into hydrocarbons in a secondary metabolic pathway.

**Hydrocarbon formation follows a secondary metabolic pathway.** To determine the physiological role of the *in vivo* CO reduction by V-nitrogenase, *A. vinelandii* cultures expressing the V-nitrogenase were prepared in the presence of 15% $^{12}CO$ or $^{13}CO$ with

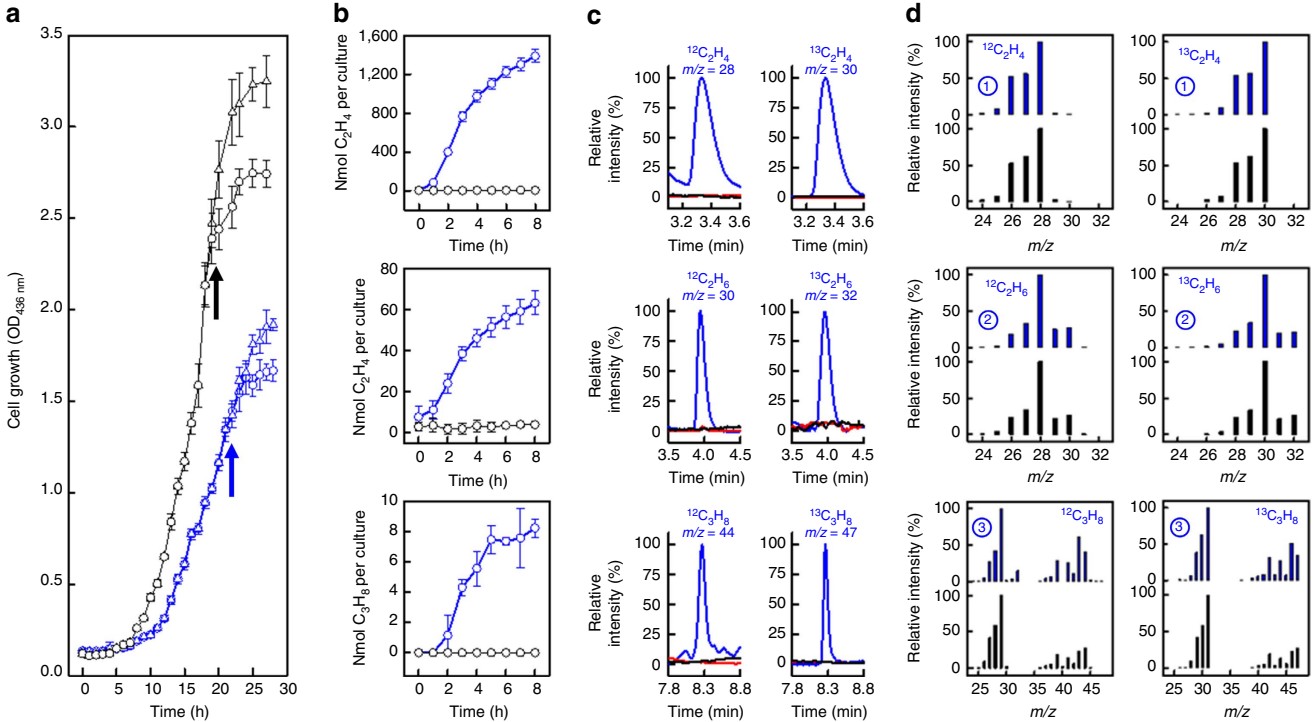

**Figure 1 | *In vivo* hydrocarbon formation from CO reduction by *A. vinelandii*.** (**a**) Growth curves of *A. vinelandii* strains expressing Mo (black)- and V (blue)-nitrogenases in the presence (open circles) and absence (open triangles) of 10% CO. The arrows indicate the time points of CO addition to the two cultures, when ammonia was depleted and nitrogenase expression was turned on. (**b**) Time-dependent formation of $C_2H_4$, $C_2H_6$ and $C_3H_8$ by 250 ml cultures of *A. vinelandii* strains expressing Mo (black)- and V (blue)-nitrogenases. The 250 ml culture expressing the V-nitrogenase yielded 1,387 nmol $C_2H_4$, 63 nmol $C_2H_6$ and 8 nmol $C_3H_8$, respectively, when 1.7 mmol CO was supplied in a 250 ml gas phase. (**c**) GC–MS analysis of $C_2H_4$ (upper), $C_2H_6$ (middle) and $C_3H_8$ (lower) formed by *A. vinelandii* strain expressing V-nitrogenase in the presence of $^{12}CO$ (left) and $^{13}CO$ (right). The products were traced in the SIM mode of GC–MS at masses indicated in the figure. The peak intensities of products generated by the culture expressing the V-nitrogenase (blue) were set at 100%. The absence of product formation when CO was supplied to controls that either expressed the Mo-nitrogenase in the absence of ammonia (black) or did not express the V-nitrogenase in the presence of ammonia (red) is noteworthy. (**d**) GC–MS fragmentation patterns of $C_2H_4$ (①, blue), $C_2H_6$ (②, blue) and $C_3H_8$ (③, blue) formed by the *A. vinelandii* strain expressing the V-nitrogenase in the presence of $^{12}CO$ (left) and $^{13}CO$ (right). The corresponding fragmentation patterns of standards are presented based on information obtained from the NIST MS database (black) (http://webbook.nist.gov). The intensities of base peaks were set at 100% in all panels. Data of cell growth and activity analysis (**a**,**b**) were obtained from three independent experiments (*n* = 6) and presented as mean ± s. d. GC–MS experiments (**c**,**d**) were conducted three times and representative results are shown.

intermittent 'relaxation' every 4 h by a short, 20 min exposure to air. This procedure was repeated 13 times, to allow isotope enrichment and, subsequently, equal amounts of the $^{12}CO$- and $^{13}CO$-treated cells were fixed, dried and analysed by nanoscale secondary ion MS (CAMECA nanoSIMS 50L instrument)[7–9]. Secondary electron images of samples incubated with $^{12}CO$ (Fig. 3a) and $^{13}CO$ (Fig. 3c) confirmed the identities of the images as those derived from cells and not from random particles, whereas the secondary ion images demonstrated nearly identical $^{13}C/^{12}C$ ratio of the $^{12}CO$-incubated (Fig. 3b) and $^{13}CO$-incubated (Fig. 3d) samples. Analysis of the $^{13}C/^{12}C$ ratios of three different regions of interest (ROI) in each sample further confirmed that the abundance of $^{13}C$ in the $^{13}CO$-incubated sample was indistinguishable from that in the $^{12}CO$-incubated sample (Fig. 3e), suggesting that the carbon of CO was not incorporated into the cellular components. Liquid chromatography–MS (LC–MS) showed indistinguishable ratios between the natural abundance (Fig. 3f, black), one $^{13}C$-incorporated (Fig. 3f, blue) and two $^{13}C$-incorporated (Fig. 3f, red) acetyl-CoA molecules in *A. vinelandii* cells incubated without CO (Fig. 3f, left), with $^{12}CO$ (Fig. 3f, middle) and with $^{13}CO$ (Fig. 3f, right) concomitant with the expression of V-nitrogenase, providing further support that CO was not used as a carbon source to generate the central metabolite, acetyl-CoA,

during cell growth. Combined results from these studies conclusively defined the *in vivo* CO reduction by V-nitrogenase as a secondary metabolic pathway for the conversion of CO into hydrocarbons.

## Discussion

Nature has developed some effective strategies for microbes to use CO as an electron and/or carbon source for cell growth[10,11]. The *in vivo* conversion of CO to alkanes/alkenes by *A. vinelandii* represents a previously unidentified strategy used by microbes to cope with CO. It has been postulated that the Earth atmosphere was rich in CO and methane ($CH_4$) in the Archean Eon[12], and that microbes living in this environment were adapted to strategies to effectively use these carbon-containing molecules, alone or together with other organisms. It is plausible, therefore, that the host of the ancestral, 'prototype nitrogenase' was well-tuned towards reducing CO to small alkenes and alkanes, and these products were then assimilated and used as the sole carbon/energy source either by the host itself or by other microbes that could not generate these products on their own. Interestingly, although nitrogenase-expressing organisms such as *A. vinelandii* seem to have lost the ability to use small hydrocarbons for cell growth during the course of evolution,

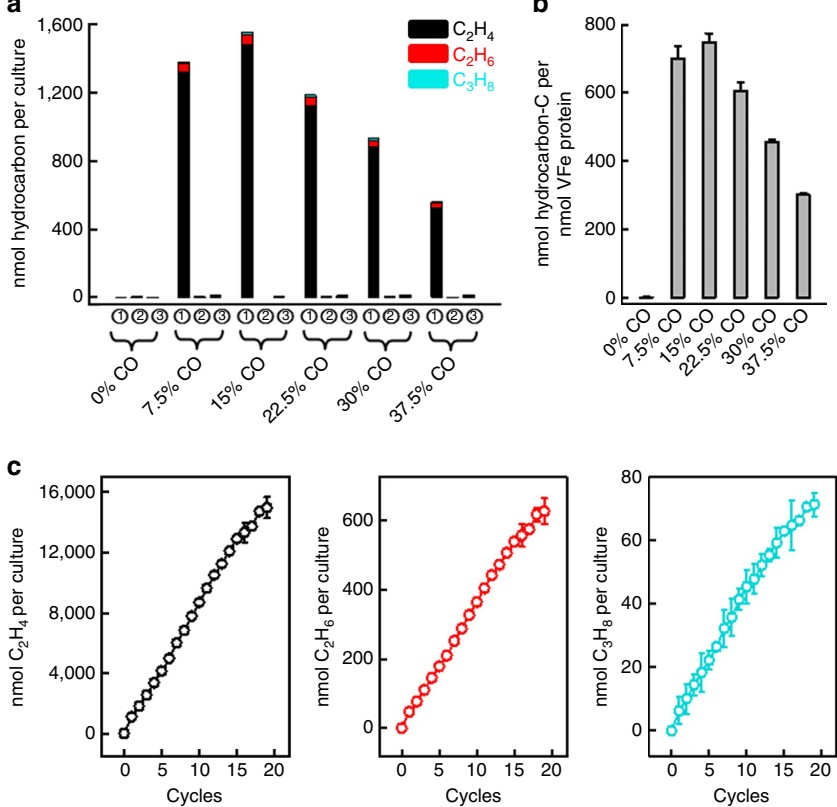

**Figure 2 | Optimization of *in vivo* hydrocarbon production from CO reduction.** (**a**) Hydrocarbon formation by 250 ml cultures of *A. vinelandii* strains carrying encoding genes for V (①,②)- and Mo (③)-nitrogenases on incubation with various CO concentrations for 8 h. CO was added on depletion of ammonia without (①,③) or with (②) supplementation of additional ammonia to the growth media. (**b**) Specific yield of *in vivo* hydrocarbon formation by V-nitrogenase upon incubation with various CO concentrations for 8 h. (**c**) Linear increases of $C_2H_4$ (left), $C_2H_6$ (middle) and $C_3H_8$ (right) formation by a 250 ml culture of the V-nitrogenase-expressing *A. vinelandii* strain every 4 h on repeated addition of CO with intermittent 20 min aeration. A 250 ml culture expressing the V-nitrogenase yielded 14,981 nmol $C_2H_4$, 626 nmol $C_2H_6$ and 71 nmol $C_3H_8$, respectively, after 20 cycles of repeated addition of 2.6 mmol CO in a 250 ml gas phase with intermittent aeration, which was equivalent to 1.2% carbon conversion. Data of activity analysis were obtained from three independent experiments ($n = 6$) and presented as mean ± s.d.

various alkane/alkene-assimilating organisms are known to exist today[13–15], implying that a nitrogenase-based, symbiotic CO-using strategy may still be in use. Moreover, the possible evolvement of nitrogenase from an enzyme specialized in processing carbon compounds into one specialized in processing nitrogen compounds establishes this enzyme as an evolutionary link between the carbon and nitrogen cycles on Earth.

Regardless of whether the ability of nitrogenase to utilize CO precedes its ability to reduce $N_2$ during evolution, the fact that CO is reduced by *A. vinelandii* to hydrocarbons as secondary metabolites gives this *in vivo* reaction a clear biotechnological advantage in maximizing the product yield and cost efficiency. Indeed, the yield of ethylene production by this reaction (1 μmol $C_2H_4$ per gram dry cell per hour) is already within the range of the existing methodologies for bio-ethylene production (0.1–3,000 μmol $C_2H_4$ per gram dry cell per hour)[16] before any optimization. Further, despite a low percentage of carbon conversion, the specific activity of this reaction (0.1 μmol hydrocarbons per gram V-nitrogenase per second) is comparable with the specific activities of the less efficient Fischer–Tropsch catalysts or those discovered at the early developmental stage of this industrial process (0.024–0.3 μmol hydrocarbons per gram catalyst per second)[17,18]. Importantly, contrary to the Fischer–Tropsch process or fermentation-based methodologies for hydrocarbon production, the *A. vinelandii*-enabled reaction is

an ambient, feedstock-free process that directly converts a toxic waste (CO) into useful hydrocarbon products. Perhaps even more excitingly, it was discovered recently that *A. vinelandii* strains expressing the respective Fe protein components of Mo- and V-nitrogenases alone were capable of *in vivo* conversion of $CO_2$ to CO (ref. 19), a process that could be coupled with the *in vivo* conversion of CO to hydrocarbons by *A. vinelandii* into a whole-cell system for hydrocarbon production from the greenhouse gas $CO_2$. Such a two-step system will circumvent the problem that a direct conversion of $CO_2$ to hydrocarbons by nitrogenase is of extremely poor efficiency[20]. In addition, both steps in this system can be optimized for improved product yield or desired product profile and, on their own or in combination, they represent attractive templates for future development of strategies that effectively recycle carbon wastes into useful chemical and fuel products.

## Methods

Unless otherwise specified, all chemicals were purchased from Fisher (Pittsburgh, PA) or Sigma-Aldrich (St Louis, MO). Natural abundance $^{12}CO$ (99.9% purity) was purchased from Praxair (Danbury, CT), whereas $^{13}CO$ (≥98% isotopic purity) was purchased from Cambridge Isotope Labs (Tewksbury, MA).

***In vivo* CO reduction by *A. vinelandii* nitrogenase.** *A. vinelandii* strains expressing the Mo- and V-nitrogenases, respectively[5,21], were grown in two 500 ml Erlenmeyer flasks, each containing 250 ml Burke's minimal medium supplemented with 2 mM ammonium acetate (NH₄OAc) and either 11 μM $Na_2MoO_4$ (in the case

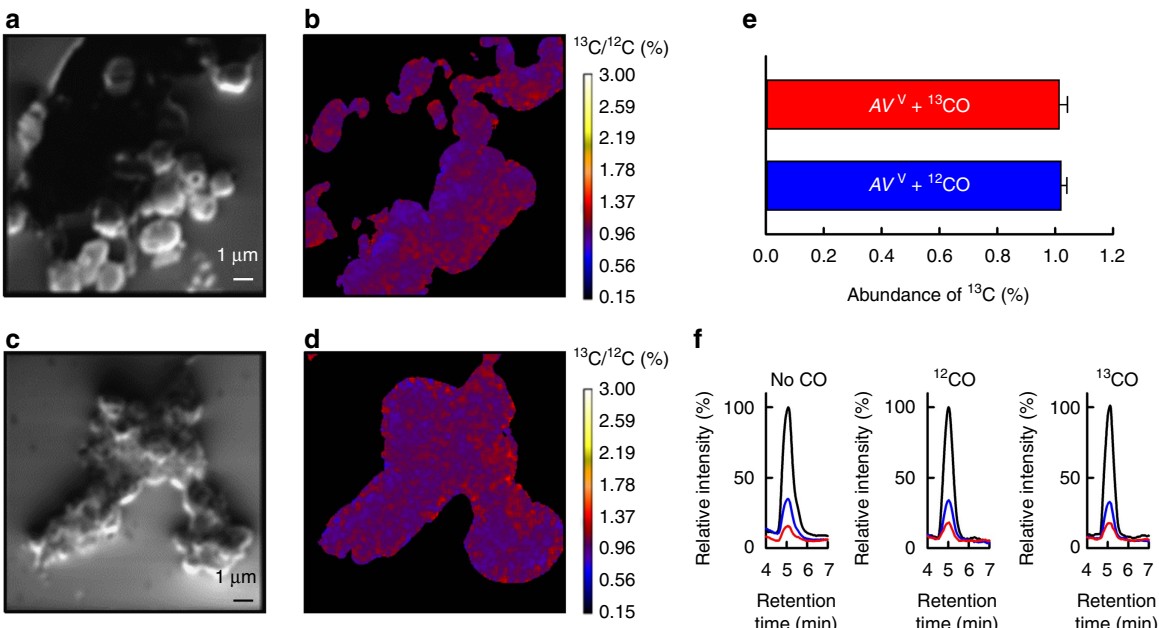

**Figure 3 | In vivo CO reduction as a secondary metabolic pathway.** (**a–d**) Secondary electron images (**a,c**) and secondary ion ($^{13}C^-$ and $^{12}C^-$) images (**b,d**) derived from nanoSIMS analysis of *A. vinelandii* cells expressing V-nitrogenase in the presence of $^{12}CO$ (**a,b**) or $^{13}CO$ (**c,d**). The nanoSIMS experiment was performed twice. Each time, data were collected at three different ROI in each sample. Representative nanoSIMS data are shown in **a–d**. (**e**) Average $^{13}C$ abundance ($^{13}C/^{12}C$ ratio) of *A. vinelandii* cells expressing V-nitrogenase in the presence of $^{12}CO$ (blue) or $^{13}CO$ (red). The average $^{13}C/^{12}C$ ratio of each sample was calculated based on data collected in three different ROI ($n = 9$). (**f**) LC–MS analysis of acetyl-CoA formed by *A. vinelandii* cells expressing V-nitrogenase in the absence (left) and presence of $^{12}CO$ (middle) or $^{13}CO$ (right). Natural abundance (black), one (blue) and two (red) $^{13}C$-substituted acetyl-CoA molecules were traced at masses 810, 811 and 812, respectively. The LC–MS experiment was performed four times. Representative results are shown in **f**. For additional nanoSIMS experiments conducted in the presence of $^{15}N_2$, see Supplementary Fig. 1. Scale bars, 1 μm (**a,c**).

of the strain expressing Mo-nitrogenase) or 30 μM $Na_3VO_4$ (in the case of the strain expressing V-nitrogenase). The cultures were grown at 30 °C with shaking at 200 r.p.m. and the growth rates were monitored by measuring cell densities at 436 nm using a Genesys 20 spectrophotometer (Spectronic, Westbury, NY). After 20 and 23 h, when $OD_{436}$ of the cultures started to plateau at ~2.4 (the strain expressing Mo-nitrogenase) and ~1.4 (the strain expressing V-nitrogenase), respectively, the Erlenmeyer flasks were capped by airtight stoppers and CO was added at a concentration of 10% to the head spaces of these flasks. Subsequently, the cultures were allowed to grow at 30 °C with shaking at 200 r.p.m. and 250 μl of headspace sample was taken from each culture at 0, 1, 2, 3, 4, 5, 6, 7 and 8 h after CO addition and examined for hydrocarbon formation. As controls, *A. vinelandii* cultures expressing the Mo- and V-nitrogenases, respectively, were prepared by the same procedure as described above, except that CO was not added when ammonia was exhausted in the growth media. Hydrocarbon products were quantified by gas chromatography with flame ionization detector (GC–FID)[4,5]. Specifically, each 250 μl headspace sample was injected onto an activated alumina column (Grace, Columbia, MD), which was held at 55 °C for 1 min, heated to 155 °C at 12.5 °C min$^{-1}$ and held at 155 °C during the course of measurement. The quantities of all products were determined by using a Scott gas mixture containing 15 p.p.m. of each hydrocarbon compound (Houston, TX). The detection limits were 10.27, 6.22, 9.73 and 2.79 nmol l$^{-1}$, respectively, for $CH_4$, $C_2H_4$, $C_2H_6$ and $C_3H_8$.

**Optimizing CO concentration for hydrocarbon formation.** *A. vinelandii* strains expressing Mo- and V-nitrogenases, respectively, were grown as described above in three sets of 500 ml Erlenmeyer flasks (six flasks per set), each containing 250 ml Burke's minimal medium supplemented with 2 mM $NH_4OAc$ and (i) and (ii) 30 μM $Na_3VO_4$ (in the case of the strain expressing V-nitrogenase), and (iii) 11 μM $Na_2MoO_4$ (in the case of the strain expressing Mo-nitrogenase). When $OD_{436}$ of the cultures reached ~2.4 (the strain expressing Mo-nitrogenase) and ~1.4 (the strain expressing V-nitrogenase), respectively, the six cultures of set (ii) were supplemented with 2 mM $NH_4OAc$. Subsequently, all flasks were capped airtight, followed by addition of CO at 0, 7.5, 15, 22.5, 30 and 37.5%, respectively, to the gas phases of the six flasks in each set of cultures. The cultures were then allowed to grow at 30 °C with shaking at 200 r.p.m. for 8 h and 250 μl of headspace sample was taken from each culture and examined by GC–FID for hydrocarbon formation (see above). The concentration of the VFe protein was determined based on the average yield of five independent purifications[21].

**Improving hydrocarbon yield with intermittent air exposure.** A 250 ml culture of *A. vinelandii* strain expressing the V-nitrogenase was grown in a 500 ml Erlenmeyer

flask as described above until the cell density started to plateau. At this point, the flask was capped airtight and CO was added at a concentration of 15% to the gas phase of this culture. The culture was then allowed to grow at 30 °C with shaking at 200 r.p.m. for 4 h before 250 μl of the headspace sample was taken and examined by GC–FID for hydrocarbon formation (see above). Following this procedure, the airtight stopper of the flask was replaced by a sterile foam plug and the culture was incubated at 30 °C with shaking at 200 r.p.m. for 20 min. Subsequently, the flask was capped airtight again, followed by addition of 15% CO to the gas phase, incubation of culture with CO for 4 h, determination of hydrocarbon formation by GC–FID and aeration of the culture for 20 min as described above. This procedure was repeated for a total of 20 times until the linear increase of hydrocarbon formation started to plateau.

**GC–MS analysis of hydrocarbon products.** *A. vinelandii* strain expressing the V-nitrogenase was grown as described above until the cell density started to plateau, when the flask was capped airtight, followed by addition of $^{13}CO$ at a concentration of 15% to the gas phase of this culture. The culture was grown for another 8 h before 250 μl headspace sample was taken and analysed by GC–MS using a Thermo Scientific Trace 1300 GC system coupled to a Thermo ISQ QD (Thermo Electron North America LLC, Madison, WI)[4,5]. Specifically, a 250 μl gas sample was injected into a split/splitless injector operated at 120 °C in in split mode, with a split ratio of 5. A 1 mm ID liner was used to optimize the sensitivity of gas separation, which was achieved on an HP-PLOT-Q capillary column (0.320 mm ID × 30 m length, Agilent Technologies, Santa Clara, CA) that was held at 40 °C for 2 min, heated to 180 °C at a rate of 10 °C min$^{-1}$ and held at 180 °C for 1 min. The carrier gas, helium, was passed through the column at a rate of 1.1 ml min$^{-1}$. The mass spectrometer was operated in electron impact ionization mode and the identities of $C_2H_4$, $C_2H_6$ and $C_3H_8$ were confirmed by comparing their masses and retention times with those of the Scott standard alkane and alkene gas mixture.

**NanoSIMS analysis of A. vinelandii cultures.** For experiments presented in Fig. 3, *A. vinelandii* strain expressing the V-nitrogenase was grown in two 250 ml Erlenmeyer flasks, each containing 100 ml Burke's minimal medium (which contained sucrose as the carbon source)[22] supplemented with 2 mM ammonium acetate and 30 μM $Na_3VO_4$. The cultures were incubated at 30 °C with shaking at 200 r.p.m. until the cell densities started to plateau. Subsequently, the Erlenmeyer flasks were capped airtight, followed by addition of 15% $^{12}CO$ and $^{13}CO$, respectively, to the gas phases of the two cultures. The cultures were allowed to undergo 13 repetitions of 4 h CO incubation with 20 min aeration, with the formation of hydrocarbons monitored throughout the process. For experiments

presented in Supplementary Fig. 1, three 100 ml cultures were prepared as described above by growing cells till the cell densities started to plateau, followed by exchange of the gas phases into gas mixtures containing (i) 65% $^{15}N_2$, 15% $^{14}N_2$ and 20% $O_2$; (ii) 65% $^{15}N_2$, 15% $^{13}CO$ and 20% $O_2$; and (iii) 65% $^{14}N_2$, 15% $^{12}CO$ and 20% $O_2$, respectively. The CO-free culture was then grown for 4 h, whereas the $^{12}CO$- and $^{13}CO$-containing cultures were allowed to undergo 13 repetitions of 4 h CO incubation with 20 min aeration. For nanoSIMS analysis, an aliquot of 250 µl of each culture (diluted to the same OD$_{436}$) was pipetted on a silicon wafer (University Wafer, Boston, MA) with a diameter of 2.5 cm. The samples were subsequently fixed with a PBS solution containing 4% formaldehyde for 1 h at room temperature, washed sequentially with PBS, 1:1 PBS/ethanol and ethanol, and dried on the wafer[9]. The secondary ion ($^{12}C^-$ and $^{13}C^-$) and secondary electron images were acquired with a CAMECA NanoSIMS 50L ion microprobe (Caltech, Pasadena, CA). A +8 keV primary Cs$^+$ beam of ~1 pA was used to raster the samples in 12 × 12, 10 × 10 or 2 × 2 µm areas. Secondary ion ($^{12}C^-$ and $^{13}C^-$) images of −8 keV were collected simultaneously with electron multiplier detectors. The interference from $^{12}CH^-$ to $^{13}C^-$ was separated with a mass resolving power of ~5,000. Ion images of 256 × 256 pixels were collected in 3 different ROI per sample, 2 or 3 frames per ROI and 15 min per frame, and processed with L'image software (http://limagesoftware.net/).

**LC–MS analysis of acetyl-CoA formation.** *A. vinelandii* strain expressing the V-nitrogenase was grown in three 250 ml batches in 500 ml Erlenmeyer flasks as described above until OD$_{436}$ of these cultures reached ~1.4. Subsequently, these cultures were harvested individually by centrifugation at 15,000 g, 4 °C for 10 min, followed by resuspension of each pellet in 250 ml Burke's minimal medium that contained no ammonia and a limited amount of sucrose (2 g l$^{-1}$; equivalent to 1/10 of that normally used in the Burke's minimal medium). The flasks were then capped airtight, followed by addition of no CO, 15% $^{12}CO$ and 15% $^{13}CO$, respectively, to the headspaces of these re-suspended cultures, continued incubation of these cultures at 30 °C with shaking at 200 r.p.m. for 18 h, harvesting of cells by centrifugation at 15,000 g, 4 °C for 10 min and storage of pellets at −80 °C. An amount of 0.5 g of each pellet was lysed by addition of 1.5 ml of 10% perchloric acid, followed by vigorous vortexing of the mixture. The cell lysate was then allowed to sit for 10 min before the pH of the solution was adjusted to 7.4 by 1 M Tris-HCl (pH 7.5). Each lysate was then filtered using Amicon Ultra 30,000 MWCO centrifugal filters (EMD Millipore, Billerica, MA) and analysed for acetyl-CoA by LC–MS analysis. Specifically, acetyl-CoA was separated by a Thermo Scientific Dionex Ultimate 3000 UHPLC system on an Acclaim 120 C18 column (4.6 × 100 mm, 5 µm particle size), which was directly coupled with a MSQ Plus single quadruple mass spectrometer (Thermo Electron North America LLC). The column was equilibrated with 100% buffer A (95%:5% H$_2$O/acetonitrile, 5 mM ammonia formate pH 7.5) for 15 min before application of samples. Each run was initiated on injection of a 100 µl sample onto the column, followed by application of an isocratic flow of 100% buffer A for 5 min, a linear gradient of 0–100% buffer B over 5 min and an isocratic flow of 100% buffer B for 10 min. To re-equilibrate the column after each run, a linear gradient of 100–0% buffer B was applied over 5 min, followed by an isocratic flow of 100% buffer A for 5 min over the column. The flow rate was kept at 0.5 ml min$^{-1}$, whereas the column was maintained at 30 °C throughout the runs. A purchased standard of acetyl-CoA (Sigma-Aldrich) was used to establish the retention time of this molecule on this column. Mass determination of different acetyl-CoA species was performed via electrospray ionization in positive ion mode with the following mass spectrometer parameters: capillary voltage, 3,000 V; sample cone voltage, 30 V; desolvation temperature, 120 °C; and source temperature, 120 °C. The masses 810 (acetyl-CoA), 811 (one $^{13}C$-incorporated acetyl-CoA) and 812 (two $^{13}C$-incorporated acetyl-CoA) were traced using the SIM mode.

**Data availability.** The authors declare that all data supporting the findings of this study are available within the article and its Supplementary Information files and from the corresponding authors upon reasonable request.

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

## Acknowledgements

This work was supported by DOE (BES) Award DE-SC0014470, funds from UC Irvine (M.W.R. and Y.H.) and a Hellman Fellowship (Y.H.).

## Author contributions

J.G.R. and C.C.L. performed experiments and analysed data. Y.H. and M.W.R. designed experiments, analysed data and wrote the paper.

## Additional information

**Competing financial interests**: The authors declare no competing financial interests.

**How to cite this article**: Rebelein, J. G. et al. The *in vivo* hydrocarbon formation by vanadium nitrogenase follows a secondary metabolic pathway. *Nat. Commun.* 7, 13641 doi: 10.1038/ncomms13641 (2016).

**Publisher's note**: 

