## [Peer Review File · Nature Communications]

Reviewer #1 (Remarks to the Author):

The authors present clear evidence that in vivo, *Azotobacter vinelandii* nitrogenase reduced CO to hydrocarbons. This is certainly of interest considering the need for better catalysts for low temperature CO processing. However, since it has already been shown that the isolated enzyme will reduce CO to hydrocarbons, this result does not appear to be surprising.

The authors show that the CO reduction products do not get incorporated into cell mass, which is disappointing. Although they speculate about combining the CO reduction process with hydrocarbon processing enzymes, it is not clear how this would work.

For example, methane monooxygenase and cytochrome p-450 both require oxygen to process hydrocarbons, which would inhibit N₂ase. Perhaps the authors could address this issue.

Reviewer #2 (Remarks to the Author):

Rebelein et al is an extension of work demonstrating the capability of purified vanadium-nitrogenase enzyme to catalyze the reduction of carbon monoxide to short chain hydrocarbons (e.g. Lee et al 2010; Hu et al 2011). The main questions addressed in this study assess whether V-nitrogenase facilitated CO reduction also occurs in vivo in V-nitrogenase expressing *Azotobacter* and whether the CO or resulting short chain hydrocarbons are directly assimilated by these organisms or excreted via a secondary metabolite pathway.

Cultured *Azotobacter* strains expressing either the vanadium or MoFe form of nitrogenase were used in a suite of experiments to assess whether this phenomenon was unique to the alternative V-nitrogenase. ¹³C-labeled carbon monoxide was used to demonstrate that CO was the carbon source by end point measurement of ¹³C enriched ethylene, ethane and propane relative to control incubations amended with unenriched CO. Analysis of the ¹³C/¹²C ratio of whole cells and cell constituents (acyl CoA) from these experiments further indicated that little if any of the hydrocarbons generated were assimilated by *Azotobacter*.

The overall experimental design is appropriate to address the questions posed in this manuscript, however the data analysis and figure display need be revisited to aid in interpretation. As presented, it is difficult for readers to independently evaluate whether the conclusions are valid from the figures (especially in regard to the lack of HC assimilation). The manuscript could be improved by revising the format for displaying the data.

Additionally, it would be helpful to know if these culture-based ¹³C experiments were replicated and, if not, were there analytical/logistical limitations that precluded running these experiments in triplicate?

Figure 1b

Demonstrating that the ^{13}C incubation produced the predicted shift in the mass of ethane, propane is critical to the conclusions and the authors should find an alternative way to plot the data that more clearly shows the mass shift relative to the parallel control incubation with unenriched CO. Perhaps include an overlay of the chromatograms? It's also unclear why this is plotted in relative abundance? The only evidence the reader has that there was truly a shift in the mass is the m/z value typed on each of the plots. At the very least, the fragmentation patterns for both CO incubations should be included in the supplement.

Specific comments

Interestingly, the V-nitrogenase of *Azotobacter vinelandii* is considerably more active than its Mo-counterpart in CO-reduction, catalyzing this reaction at a rate of 16 nmol reduced carbon/nmol protein/min⁵.

Include the reaction rate for the MoFe protein as well.

Pg 6: why would there be dirt particles in your cultures?

Pg. 6: you can't do phase contrast microscopy with a 'regular electron microscope'.

Pg. 6: In the discussion of whether the cells were capable of assimilating HC, it would be useful to do a simple back of the envelope calculation based on rates of HC production to estimate the theoretical amount of CO derived C that would need to be assimilated during your experiment in order to detect a shift in the $^{13}\text{C}/^{12}\text{C}$ of biomass. Are the azotobacter using acetate from the ammonium acetate added as their carbon source? Is there any other carbon added to Burke's minimal media, or are they both nitrogen and carbon starved upon the addition of CO? The OD growth curves in figure 1a indicate growth is stimulated by the addition of CO, but not clear what the cells are using if not assimilating the CO.

Pg 6: nanoscale secondary ion mass spectrometry (CAMECA nanoSIMS 50L instrument).

Pg 6: ratio values should be reported as fractional abundance or, in the case where there is little to no ^{13}C enrichment, converted to delta notation. Also include information on the ^{13}C standard used.

Pg 6: what is meant by '3 random sections'? Does this mean 3 different raster areas? different frames/ cycle #'s?

Pg 7 'exist in present days' exist today?

Pg7: "Combined results from these studies conclusively defined the in vivo CO reduction by Vnitrogenase as a novel, secondary metabolic pathway specializing in the conversion of CO into hydrocarbons".

This seems to imply that the V-nitrogenase is not primarily functioning in the fixation of N_2 ? Not sure the experimental evidence presented here conclusively shows that this isoform of the enzyme

specializes in CO to hydrocarbon? If you had additionally supplied $^{15}\text{N}_2$, would you have observed N_2 fixation and assimilation in addition to CO reduction?

Pg 7 bottom of page. "Evolution of nitrogenase from carbon processing to nitrogen fixation".

Why assume the nitrogenase enzyme evolved first to reduce carbon and then later became optimized for nitrogen fixation? Recent phylogenetic based evidence by Boyd and others indicates that the MoFe form of nitrogenase appears to be ancestral to the Fe and V forms.

METHODS SECTION

Pg9: more information and references should be included about how you determine whether this is a vanadium or molybdenum expressing Azotobacter culture- is this exclusively controlled by availability of V or Mo in culture, is this a wild type strain with all 3 active forms of nitrogenase or is this a modified Azotobacter that only has one functional form of the nitrogenase enzyme?

Pg 9: $^{13}\text{CO}_2$? Assuming this is a typo

Pg 10: The concentration of the VFe protein was determined based on the average yield of five independent purifications

How was this done? If this is a standard protocol, it's important to cite the reference.

Pg10: did you detect any CH_4 production in your experiments?

Pg12-13- It's not clear based on the description of the culture experiment whether there were 3 independent batches (3x 250 ml replicates) for each treatment in the ^{13}C labeling experiment, or a single bottle of 250 ml for each treatment.

Did you measure acetyl CoA or acyl coA? The heading in the text on page 12 indicates acyl CoA

Pg 12: were the cultures used for the nanoSIMS analysis the same cultures used in the HC production from figure 1? It would help to have a statement at the start of the methods how many independent culture incubations were used and which results are from the same set of ^{13}C experiments.

Pg 12: Why were the cultures first dried on the slide and then fixed with paraformaldehyde? This is not the conventional protocol. Typically paraformaldehyde fixation is added immediately to the liquid culture to maximize cell preservation? Fixing with paraformaldehyde after drying the cells on the wafer doesn't make sense.

Pg 12: reference for L'image software?

more information needs to be included in the figure captions.

Figure 1a: The azotobacter cultures clearly are showing enhanced growth in the V-nitrogenase cultures supplied with CO, is the secondary HC metabolite production stimulating enhanced respiration of supplied organic carbon in the cultures?

Figure 1 legend: what do the error bars represent in panel a and b? biological replicates, technical replicates?

Figure 3a: when you are dealing with potentially low levels of ^{13}C incorporation, it is important to convert to delta notation. The color scale should at the very least be set to the same minimum and maximum value for your ^{12}C experiments in panels a/b and ^{13}C cells in c/d. It would also be useful to have information about what the natural abundance $^{13}\text{C}/^{12}\text{C}$ ratio is for your known isotopic standard.

Figure 3e: not sure what the point of this is. Should really use a more conventional way of displaying $\delta^{13}\text{C}$ data.

Reviewer #3 (Remarks to the Author):

This paper describes CO conversion to hydrocarbons (ethylene mainly) by *Azotobacter vinelandii*. As this is an original, novel, secondary metabolic reaction, it might be worth publishing.

Some comments on data and methodology and their possible improvement are given hereafter.

Pg. 3: would be interesting briefly explaining where toxic CO could be obtained from, in order to recycle it as useful product as suggested by the authors.

Pg. 3: If the rates of both nitrogenases are to be compared, then both values should be given rather than only one (16 nmolC/nmol/min); otherwise the "higher" activity cannot be quantified by the reader.

Pg. 4: The initial CO concentration is expressed as %, while the amounts C_2H_4 , C_2H_6 and C_3H_6 are expressed as nmol. How much was the actual concentration (nmol/volume)? How much CO was converted to hydrocarbons (conversion yield and efficiency)??

Pg. 5: any idea about which toxic waste products or inhibitors might accumulate, if any?

Pg. 5, last line: would be worth calculating how much CO (%) was converted to hydrocarbons.

Pg. 9: Where were the *A. vinelandii* strains expressing MO- V- nitrogenases obtained from (culture collection?, reference?, ...). Without such information, it is impossible to double check or repeat the experiment.

Pg. 9 ("... added at a concentration of 10% to the headspace ..."): what was the composition of the gas phase before CO addition (air?)? How was the addition of 10% CO measured? How was such

percentage confirmed?, ... I guess this would mean that the gas phase contains 25 mL CO and 225 mL of another gas (which one? ... (presumably air?)). Please explain/confirm!

How much was the total gas pressure? Did the gas phase initially contain 250 mL gas (air)? ... if 10% CO was added, does this mean that there was some overpressure? ... Quantify!

Pg. 10 (2nd line): units used are not clear to me ... If hydrocarbons are analyzed by GC/FID as gas samples, GC results are expected to be "nmol" or "nmol/l" rather than "nmol/umol protein". Please explain and clarify!

Pg. 10 ("...addition of CO at 0, 7.5%, ..."). How were such percentages calculated or estimated?

Pg. 10 (bottom): How much was the total gas pressure in the flasks?

Did such pressure vary as a result of CO consumption?

The conclusions can be considered to be clear, valid and reliable.

References are appropriate.

Point-by-Point Response to Reviewers' Comments

Referee #1:

Comment 1: The authors show that the CO reduction products do not get incorporated into cell mass, which is disappointing. Although they speculate about combining the CO reduction process with hydrocarbon processing enzymes, it is not clear how this would work. For example, methane monooxygenase and cytochrome p-450 both require oxygen to process hydrocarbons, which would inhibit N₂ase. Perhaps the authors could address this issue.

Response: We did not speculate about combining the CO reduction process with hydrocarbon processing enzymes; rather, we suggest the possibility to combine the *in vivo* reduction of CO₂ to CO by the Fe protein of nitrogenase (please see the attached, related manuscript that is under revision elsewhere) with the *in vivo* reduction of CO to hydrocarbons by the V-nitrogenase in *A. vinelandii* into a two-step, whole-cell approach to ambient conversion of the greenhouse gas CO₂ into useful hydrocarbon products. In this light, the fact that the hydrocarbon products of CO reduction by the Fe protein are not incorporated into cell mass using a GC coupled with a methanizer. The reviewer is correct: the gas phase containedvel it establishes this reaction as a novel secondary metabolic pathway, which, as demonstrated by other well-known secondary metabolic pathways (such as those specialized in the production of antibiotics), is significant both from a theoretical standpoint and in a practical vein. Although the *in vitro* CO reduction by V-nitrogenase was discovered earlier, the ability of this enzyme to catalyze the *in vivo* CO reduction to hydrocarbons in *A. vinelandii* is a big surprise, given the known effect of CO in suppressing respiration and inhibiting cell growth, as well as other potential complications of this reaction that may arise from various cellular components and metabolic pathways. Our initial success in improving the product yield via repeated cycling between CO and air in the gas phase marks the first step toward exploring the potential of this system as a template for future development of whole-cell approaches to ambient CO₂/CO conversion and, as such, is of significance from the perspectives of both environmental wellness and energy production.

Referee #2:

General Comment 1: The overall experimental design is appropriate to address the questions posed in this manuscript, however the data analysis and figure display need be revisited to aid in interpretation. As presented, it is difficult for readers to independently evaluate whether the conclusions are valid from the figures (especially in regard to the lack of HC assimilation). The manuscript could be improved by revising the format for displaying the data.

Response: The reviewer's point is well taken and we have revised the manuscript as suggested by the reviewer to improve the clarity of data presentation (please see below for details).

General Comment 2: Additionally, it would be helpful to know if these culture-based ^{13}C experiments were replicated and, if not, were there analytical/logistical limitations that precluded running these experiments in triplicate?

Response: The culture-based ^{13}C experiments were conducted three times (each with two technical replicates) and the standard deviations of these data were indicated as error bars in Fig. 1. We have included information regarding the repetition of experiments in all figure legends for improved clarity.

General Comment 3: Figure 1b - Demonstrating that the ^{13}CO incubation produced the predicted shift in the mass of ethane, propane is critical to the conclusions and the authors should find an alternative way to plot the data that more clearly shows the mass shift relative to the parallel control incubation with unenriched CO . Perhaps include an overlay of the chromatograms? It's also unclear why this is plotted in relative abundance? The only evidence the reader has that there was truly a shift in the mass is the m/z value typed on each of the plots. At the very least, the fragmentation patterns for both CO incubations should be included in the supplement.

Response: The reason why we originally presented the data as individual peaks and plotted the data in relative abundance is because we used the SIM mode of GC-MS to trace the products at their expected masses as we did in our previous work along this line [*e.g.*, Lee, Hu & Ribbe, *Science* **329**, 642 (2010); Hu, Lee & Ribbe, *Science* **333**, 753-755 (2011)]. However, we recognize the reviewer's concern here and have included the fragmentation patterns of products generated upon incubation of cultures with ^{12}CO and ^{13}CO (please see Fig. 1d of the revised manuscript). In addition, we have included masses traced in the SIM mode of two control cultures that do not generate hydrocarbon products: a culture in which the expression of Mo-nitrogenase is turned on in the absence of ammonia (Fig. 1c, black trace); and another culture in which the expression of the V-nitrogenase is suppressed in the presence of ammonia (Fig. 1c, red trace). Together, these data conclusively show that CO is the source of hydrocarbon products generated by the cultures expressing V-nitrogenase.

Specific Comment 1: Interestingly, the V-nitrogenase of *Azotobacter vinelandii*⁶ is considerably more active than its Mo-counterpart in CO -reduction, catalyzing this reaction at a rate of 16 nmol reduced carbon/nmol protein/min⁵. Include the reaction rate for the MoFe protein as well.

Response: We have included the reaction rate of the MoFe protein (0.02 nmol reduced carbon/nmol protein/min; please see p. 3 of the revised manuscript).

Specific Comment 2: Pg. 6: why would there be dirt particles in your cultures?

Response: We used "dirt particles" as an example of the "random particles" that are not derived from the cells (or cultures). To avoid confusion, we have removed this phrase from the text.

Specific Comment 3: Pg. 6: you can't do phase contrast microscopy with a 'regular electron microscope'.

Response: We have modified the sentence as “Secondary electron images of samples incubated with ^{12}C (Fig. 3a) and ^{13}C (Fig. 3c) confirmed the identities of the images as those derived from cells and not from random particles”.

Specific Comment 4: Pg. 6: In the discussion of whether the cells were capable of assimilating HC, it would be useful to do a simple back of the envelope calculation based on rates of HC production to estimate the theoretical amount of CO derived C that would need to be assimilated during your experiment in order to detect a shift in the $^{13}\text{C}/^{12}\text{C}$ of biomass. Are the azotobacter using acetate from the ammonium acetate added as their carbon source? Is there any other carbon added to Burke's minimal media, or are they both nitrogen and carbon starved upon the addition of CO? The OD growth curves in figure 1a indicate growth is stimulated by the addition of CO, but not clear what the cells are using if not assimilating the CO.

Response: We realized that the concern of the reviewer about CO/HC assimilation was caused by a mistake in the legend of Fig. 1a. This was an oversight on our part, and we have corrected the legend as follows (p. 18): “...in the presence (o) and absence (Δ) of 10% CO.” As shown in Fig. 1a, addition of CO suppresses (and not activate) the growth of cultures expressing the Mo- (black circles) and V- (blue circles) nitrogenases, which is consistent with the known effect of CO on respiration and cell growth. The carbon source used for cell growth is sucrose. Our data conclusively show that there is no incorporation of CO-derived carbon into the cell mass (Fig. 3). In the absence of assimilated hydrocarbon products, there is no basis to assume that there is carbon incorporation accompanying the *in vivo* CO reduction by V-nitrogenase or use the rate at which hydrocarbon products are formed/released to derive a yield of theoretical carbon incorporation.

Specific Comment 5: Pg. 6: nanoscale secondary ion mass spectrometry (CAMECA nanoSIMS 50L instrument).

Response: The phrase in question has been modified as per the reviewer's suggestion (p. 6).

Specific Comment 6: Pg. 6: ratio values should be reported as fractional abundance or, in the case where there is little to no ^{13}C enrichment, converted to delta notation. Also include information on the ^{13}C standard used.

Response: The ratio values have been converted to delta notation as per the reviewer's suggestion (Fig. 3; Supplementary Fig. 1). The standard used for calculations of $\delta^{13}\text{C}$ and $\delta^{15}\text{N}$ values were: a $^{13}\text{C}/^{12}\text{C}$ isotope ratio of 0.011237 (Ref. 24) and a $^{15}\text{N}/^{14}\text{N}$ isotope ratio of 0.003677 (Refs. 25-27), respectively. This information is incorporated in the revised text (p. 13).

Specific Comment 7: Pg. 6: what is meant by '3 random sections'? Does this mean 3 different raster areas? different frames/ cycle #'s?

Response: The phrase “3 random sections” refers to 3 different regions of interest (ROI) in each sample, and 3 frames were collected for each ROI. This information has been updated (p. 13) as we have included new nanoSIMS data in the revised manuscript (Supplementary Fig. 1).

Specific Comment 8: Pg. 7 'exist in present days' exist today?

Response: The phrase in question has been modified in the revised text (p. 7).

Specific Comment 9: Pg. 7: "Combined results from these studies conclusively defined the in vivo CO reduction by Vnitrogenase as a novel, secondary metabolic pathway specializing in the conversion of CO into hydrocarbons". This seems to imply that the V-nitrogenase is not primarily functioning in the fixation of N₂? Not sure the experimental evidence presented here conclusively shows that this isoform of the enzyme specializes in CO to hydrocarbon? If you had additionally supplied ¹⁵N₂, would you have observed N₂ fixation and assimilation in addition to CO reduction?

Response: We clearly stated that the “novel, secondary metabolic pathway” was specialized in the conversion of CO to hydrocarbons and did not imply that the V-nitrogenase did not primarily function in N₂ fixation. However, to avoid confusion, the sentence has been modified as “...a novel, secondary metabolic pathway for the conversion of CO into hydrocarbons”. As expected from the known effect of CO in inhibiting N₂ fixation, we observed minimum N assimilation when ¹⁵N₂ was supplied in addition to CO under our reaction conditions (see Supplementary Fig. 1).

Specific Comment 10: Pg. 7 bottom of page. "Evolvement of nitrogenase from carbon processing to nitrogen fixation". Why assume the nitrogenase enzyme evolved first to reduce carbon and then later became optimized for nitrogen fixation? Recent phylogenetic based evidence by Boyd and others indicates that the MoFe form of nitrogenase appears to be ancestral to the Fe and V forms.

Response: The statement is intended as a point of discussion based on our observation that V-nitrogenase retains the CO-reducing activity (as a secondary metabolic pathway) at a certain percentage of its N₂-reducing activity, and our suggestion that nitrogenase evolved from a carbon-processing enzyme to one that fixes nitrogen seems reasonable given the change of gas compositions in the earth atmosphere during evolution. As such, our statement neither supports nor contradicts the phylogenetic-based prediction of the sequence in which different types of nitrogenases appeared in time.

Comment 1 regarding method section: Pg. 9: more information and references should be included about how you determine whether this is a vanadium or moly expressing Azotobacter culture- is this exclusively controlled by availability of V or Mo in culture, is this a wild type

strain with all 3 active forms of nitrogenase or is this a modified Azotobacter that only has one functional form of the nitrogenase enzyme?

Response: We have included two references with information of the strains used for the expression of Mo- (YM13A) and V- (YM68A) nitrogenases (Refs. 5, 21) in the revised manuscript. It has been well-established that (1) the presence of Mo in the growth medium suppresses the expression of V- and Fe-only nitrogenases, thereby only permitting expression of the Mo-nitrogenase in YM13A when Mo is supplemented in the growth medium; and (2) deletion of genes encoding the MoFe protein and the Mo transporter (ModA) in YM68A permits expression of only the V-nitrogenase when V is supplemented in the growth medium.

Comment 2 regarding method section: Pg. 9: $^{13}\text{CO}_2$? Assuming this is a typo

Response: This typo has been corrected as ^{13}CO .

Comment 3 regarding method section: Pg. 10: The concentration of the VFe protein was determined based on the average yield of five independent purifications. How was this done? If this is a standard protocol, it's important to cite the reference.

Response: A reference describing this protocol, which has been used routinely to determine the concentration of the VFe protein, has been included in the revised manuscript (Ref. 21).

Comment 4 regarding method section: Pg. 10: did you detect any CH_4 production in your experiments?

Response: No.

Comment 5 regarding method section: Pg. 12-13- It's not clear based on the description of the culture experiment whether there were 3 independent batches (3x 250 ml replicates) for each treatment in the ^{13}CO labeling experiment, or a single bottle of 250 ml for each treatment.

Response: As stated in the original text, the culture experiment was conducted with 3 independent batches of 250 mL cultures. In addition, the activity of each culture was measured in duplicates. We have included information regarding the replication of experiments in the revised manuscript (p. 10; p. 18) for improved clarity.

Comment 6 regarding method section: Did you measure acetyl CoA or acyl coA? The heading in the text on page 12 indicates acyl CoA

Response: We measured acetyl CoA. The typo has been corrected in the revised text (p. 13).

Comment 7 regarding method section: Pg. 12: were the cultures used for the nanoSIMS analysis the same cultures used in the HC production from figure 1? It would help to have a

statement at the start of the methods how many independent culture incubations were used and which results are from the same set of ^{13}C experiments.

Response: As described in the Methods section, samples for the nano SIMS analysis (Fig. 3) were prepared by repeated additions of ^{13}CO and intermittent aeration (p. 12); and samples for the activity analysis (Fig. 1) were prepared the same way but without repeated additions of CO (p. 9). Information regarding the repetition of experiments is now included in all figure legends.

Comment 8 regarding method section: Pg. 12: Why were the cultures first dried on the slide and then fixed with paraformaldehyde? This is not the conventional protocol. Typically paraformaldehyde fixation is added immediately to the liquid culture to maximize cell preservation? Fixing with paraformaldehyde after drying the cells on the wafer doesn't make sense.

Response: The sentence in question has been modified in the revised text (p. 12).

Comment 9 regarding method section: Pg. 12: reference for L'image software?

Response: L'image software was written by Larry R. Nittler, copyright of the Carnegie Institution of Washington. There is no reference for this software, but we have included a link to the related webpage in the revised manuscript (p. 13).

Comment 1 regarding figure captions: More information needs to be included in the figure captions. Figure 1a: The azotobacter cultures clearly are showing enhanced growth in the V-nitrogenase cultures supplied with CO, is the secondary HC metabolite production stimulating enhanced respiration of supplied organic carbon in the cultures?

Response: We have corrected the legend as follows (p. 18): "...in the presence (o) and absence (Δ) of 10% CO." Please see our response to Specific Comment 4 above.

Comment 2 regarding figure captions: Figure 1 legend: what do the error bars represent in panel a and b? biological replicates, technical replicates?

Response: The error bars in panels a and b of Fig. 1 were derived from 3 biological replicates (i.e., 3 different samples), each with 2 technical replicates (i.e., each sample was measured twice). This information has been included in the figure legend for improved clarity.

Comment 3 regarding figure captions: Figure 3a: when you are dealing with potentially low levels of ^{13}C incorporation, it is important to convert to delta notation. The color scale should at the very least be set to the same minimum and maximum value for your ^{12}CO experiments in panels a/b and ^{13}CO cells in c/d. It would also be useful to have information about what the natural abundance $^{13}\text{C}/^{12}\text{C}$ ratio is for your known isotopic standard.

Response: The data have been converted to delta notation, and the color scale has been set as per the reviewer's suggestion in panels a/b and c/d. The standard used for calculations of $\delta^{13}\text{C}$ value was a $^{13}\text{C}/^{12}\text{C}$ isotope ratio of 0.011237 (Ref. 24)

Comment 4 regarding figure captions: Figure 3e: not sure what the point of this is. Should really use a more conventional way of displaying d13C data.

Response: This figure is now presented in delta notation.

Referee #3:

Comment 1: Pg. 3: would be interesting briefly explaining where toxic CO could be obtained from, in order to recycle it as useful product as suggested by the authors.

Response: We have included a sentence stating that the toxic CO is found in the waste gas of steel, PVC and ferroalloys industries in the revised manuscript (p. 3).

Comment 2: Pg. 3: If the rates of both nitrogenases are to be compared, then both values should be given rather than only one (16 nmolC/nmol/min); otherwise the "higher" activity cannot be quantified by the reader.

Response: The reviewer's point is well taken, and the reaction rate of the Mo-nitrogenase in CO reduction (0.02 nmol reduced carbon/nmol protein/min) is now included in the revised manuscript (please see p. 3).

Comment 3: Pg. 4: The initial CO concentration is expressed as %, while the amounts C_2H_4 , C_2H_6 and C_3H_6 are expressed as nmol. How much was the actual concentration (nmol/volume)? How much CO was converted to hydrocarbons (conversion yield and efficiency)??

Response: We have included information of the total volume of gas phase (in mL), the amount of CO supplied (in mmol) and the amounts of products generated (in nmol) per 250 mL culture in the legend of Fig. 1. We have also included the total yields of products (in nmol) and the yield of carbon conversion (in %) upon repeated addition of CO in the legend of Fig. 2c.

Comment 4: Pg. 5: any idea about which toxic waste products or inhibitors might accumulate, if any?

Response: Unfortunately, at this point, we do not know which toxic waste products or inhibitors might accumulate in the *in vivo* reduction of CO.

Comment 5: Pg. 5, last line: would be worth calculating how much CO (%) was converted to hydrocarbons.

Response: We have included this information in the legend of Fig. 2c. In addition, we have included a brief discussion of how the specific activity of the whole-cell CO reduction by *A. vinelandii* compares with those of the Fischer-Tropsch catalysts as follows in the revised text (p. 8): "Further, despite a low percentage of carbon conversion, the specific activity of this reaction (0.1 μmol hydrocarbons/g V-nitrogenase/s) is comparable with the specific activities of the less efficient Fischer-Tropsch catalysts or those discovered at the early developmental stage of this industrial process (0.024-0.3 μmol hydrocarbons/g catalyst/s)^{18,19}."

Comment 6: Pg. 9: Where were the *A. vinelandii* strains expressing Mo- and V- nitrogenases obtained from (culture collection?, reference?, ...). Without such information, it is impossible to double check or repeat the experiment.

Response: We have included two references with information of the strains used for the expression of Mo- (YM13A) and V- (YM68A) nitrogenases (Refs. 5, 21) in the revised manuscript.

Comment 7: Pg. 9 ("... added at a concentration of 10% to the headspace ..."): what was the composition of the gas phase before CO addition (air?)? How was the addition of 10% CO measured? How was such percentage confirmed?, ... I guess this would mean that the gas phase contains 25 mL CO and 225 mL of another gas (which one? ... (presumably air?)). Please explain/confirm!

Response: The composition of the gas phase before CO addition was air. Addition of 10% CO was done by using two syringes to substitute CO for air, and the percentage of CO after gas exchange was confirmed by GC-FID measurement using a GC coupled with a methanizer. The reviewer is correct: the gas phase contained 25 mL CO and 225 mL air for the sample containing 10% CO.

Comment 8: How much was the total gas pressure? Did the gas phase initially contain 250 mL gas (air?)? ... if 10% CO was added, does this mean that there was some overpressure? ... Quantify!

Response: The total gas pressure remained 1 atm upon CO addition, and there was no over pressure in the gas phase (please see above).

Comment 9: Pg. 10 (2nd line): units used are not clear to me ... If hydrocarbons are analyzed by GC/FID as gas samples, GC results are expected to be "nmol" or "nmol/l" rather than "nmol/ μmol protein". Please explain and clarify!

Response: We have modified the unit as nmol/L (p. 10).

Comment 10: Pg. 10 ("...addition of CO at 0, 7.5%, ..."). How were such %ages calculated or estimated?

Response: CO was added at various amounts in the gas phase of the culture and confirmed for final concentration as described above (please see our response to Comment 7 above).

Comment 11: Pg. 10 (bottom): How much was the total gas pressure in the flasks? Did such pressure vary as a result of CO consumption?

Response: The total gas pressure in the flasks is 1 atm. We did not detect pressure variation during the course of experiment.

Reviewer #2 (Remarks to the Author):

The responses to the reviewer's comments and edits to the manuscript are for the most part reasonable and the overall submission is much improved.

I have two specific points that still require attention:

1) The paragraph discussing evolutionary implications and links between the C and N cycles is interesting, however I did not find the author's reasoning for implicitly stating the evolutionary order of substrate preference by nitrogenase from C to N (lines 167-169) to be particularly compelling given the geochemical, isotopic and theoretical literature regarding N₂ availability during the Archean (for an example see Stuken et al 2015, Nature). Yes, reduced hydrocarbons and CO were also thought to be more prevalent relative to the atm today, but to my knowledge I do not think there is any published evidence that suggests N₂ was limiting in the archean. It is certainly valid for the authors to suggest a possible link between C and N cycling through nitrogenase, but the sentence should be reworded to avoid unsupported statements of directionality of the process (or alternatively back this statement with independent published evidence).

2) While the authors did change the nanoSIMS ¹³C/¹²C ion image to delta notation, the range on the scale displayed is inappropriate for the target analyzed, spanning from -900‰ to +900‰. A more appropriate way to display this type of data is to set the minimum value on the color scale (black) to natural abundance or just slightly below if you want to visualize the cells in the ratio image. I was surprised that the average d¹³C values reported for *A. vinelandii* in panel e is -80‰. Do you have an explanation for this? I believe the reported results in the context of showing a lack of enrichment between the ¹³CO and ¹²CO incubations, but the absolute d¹³C values reported seem far off of the d¹³C values expected for heterotrophic bacteria. If the nanoSIMS measured d¹³C values are not well calibrated to a known standard, you might instead consider reporting this in a comparative sense rather than absolute. Instead, showing a comparison of d¹³C of cells from the ¹³CO incubation against values from the ¹²CO incubation. I would also suggest adding a 3rd panel in the nanoSIMS image display that shows the total cell biomass using either the ¹²C- or ¹²C¹⁴N- ion image in addition to the ratio image.

Point-by-Point Response to Referee's Comments

Referee #2:

Comment 1: The paragraph discussing evolutionary implications and links between the C and N cycles is interesting, however I did not find the author's reasoning for implicitly stating the evolutionary order of substrate preference by nitrogenase from C to N (lines 167-169) to be particularly compelling given the geochemical, isotopic and theoretical literature regarding N₂ availability during the Archean (for an example see Stuken et al 2015, Nature). Yes, reduced hydrocarbons and CO were also thought to be more prevalent relative to the atm today, but to my knowledge I do not think there is any published evidence that suggests N₂ was limiting in the archaean. It is certainly valid for the authors to suggest a possible link between C and N cycling through nitrogenase, but the sentence should be reworded to avoid unsupported statements of directionality of the process (or alternatively back this statement with independent published evidence).

Response: The lead sentence of the paragraph following the sentence in question has been modified to address the referee's concern: "Regardless of whether the ability of nitrogenase to utilize CO precedes its ability to reduce N₂ during evolution ..."

Comment 2: While the authors did change the nanoSIMS ¹³C/¹²C ion image to delta notation, the range on the scale displayed is inappropriate for the target analyzed, spanning from -900‰ to +900‰. A more appropriate way to display this type of data is to set the minimum value on the color scale (black) to natural abundance or just slightly below if you want to visualize the cells in the ratio image. I was surprised that the average d¹³C values reported for *A. vinelandii* in panel e is -80‰. Do you have an explanation for this? I believe the reported results in the context of showing a lack of enrichment between the ¹³CO and ¹²CO incubations, but the absolute d¹³C values reported seem far off of the d¹³C values expected for heterotrophic bacteria. If the nanoSIMS measured d¹³C values are not well calibrated to a known standard, you might instead consider reporting this in a comparative sense rather than absolute. Instead, showing a comparison of d¹³C of cells from the ¹³CO incubation against values from the ¹²CO incubation. I would also suggest adding a 3rd panel in the nanoSIMS image display that shows the total cell biomass using either the ¹²C- or ¹²C¹⁴N- ion image in addition to the ratio image.

Response: We agree with the referee that it makes more sense to report the nanoSIMS data in a relative scale. Therefore, we changed the scale of nanoSIMS ¹³C/¹²C ion image back to the ¹³C/¹²C ratio we used in the original submission. We did not include a third panel displaying the total cell biomass on the basis of ¹²C data. These ¹²C data were already used to calculate the ¹³C/¹²C ratios of the samples and, alone, they do not provide further information relevant to the ¹³C enrichment in these samples.